

# Deep learning-driven dyslexia detection model using multi-modality data

Yazeed Alkhurayyif[1] and Abdul Rahaman Wahab Sait[2]

[1] Department of Computer Science, Shaqra University, Shaqra, Saudi Arabia
[2] Department of Documents and Archive, Center of Documents and Administrative Communication, King Faisal University, Al-Ahsa, Saudi Arabia

## ABSTRACT

**Background**. Dyslexia is a neurological disorder that affects an individual's language processing abilities. Early care and intervention can help dyslexic individuals succeed academically and socially. Recent developments in deep learning (DL) approaches motivate researchers to build dyslexia detection models (DDMs). DL approaches facilitate the integration of multi-modality data. However, there are few multi-modality-based DDMs.

**Methods**. In this study, the authors built a DL-based DDM using multi-modality data. A squeeze and excitation (SE) integrated MobileNet V3 model, self-attention mechanisms (SA) based EfficientNet B7 model, and early stopping and SA-based Bi-directional long short-term memory (Bi-LSTM) models were developed to extract features from magnetic resonance imaging (MRI), functional MRI, and electroencephalography (EEG) data. In addition, the authors fine-tuned the LightGBM model using the Hyperband optimization technique to detect dyslexia using the extracted features. Three datasets containing FMRI, MRI, and EEG data were used to evaluate the performance of the proposed DDM.

**Results**. The findings supported the significance of the proposed DDM in detecting dyslexia with limited computational resources. The proposed model outperformed the existing DDMs by producing an optimal accuracy of 98.9%, 98.6%, and 98.8% for the FMRI, MRI, and EEG datasets, respectively. Healthcare centers and educational institutions can benefit from the proposed model to identify dyslexia in the initial stages. The interpretability of the proposed model can be improved by integrating vision transformers-based feature extraction.

# INTRODUCTION

Dyslexia is a neurological disorder that affects wider age groups across the globe (*Ahire et al., 2023*). Typically, dyslexia disrupts phonological processing, which impacts letter-to-sound mapping and word recognition (*Perera, Shiratuddin & Wong, 2018*). Dyslexia may have a substantial influence on academic achievement, especially in the areas of reading, spelling, and writing. Dyslexic children have challenges comprehending the alphabet and other fundamental reading skills (*Perera, Shiratuddin & Wong, 2018*). Dyslexic adults may

Corresponding author
Yazeed Alkhurayyif,
yalkhurayyif@su.edu.sa

struggle in higher education or professional contexts that demand excellent reading and writing skills (*Perera, Shiratuddin & Wong, 2018*). Dyslexic individuals (DIs) may struggle to believe in their unique abilities and feel less motivated to learn than their normal classmates. To overcome the challenges, the DI employs assistive technologies. Despite their potential efficacy, these technologies may be tedious and time-consuming, leaving the DIs exhausted and frustrated (*Kaisar & Chowdhury, 2022*). Dyslexia detection (DD) offers valuable insights into the neurobiological mechanisms of dyslexia. It can lead to customized educational interventions that provide the learning demands of DIs. Educators can support DIs in achieving their goals. Moreover, DD can prevent secondary consequences and empowers DIs and their families.

Functional MRI (FMRI), and electroencephalography (EEG) are widely used to detect and comprehend dyslexia (*Kaisar & Chowdhury, 2022*). With FMRI, researchers can track the brain's blood flow in real time, revealing actively engaged regions in completing various tasks (*Usman et al., 2021*). Researchers discovered dyslexia-related brain regions by comparing dyslexic and non-dyslexic neural activity during reading tests (*Jan & Khan, 2023*). These discoveries may lead to a deeper understanding of the neurological processes behind dyslexia. Physicians can monitor the brain's electrical activity using the EEG's high temporal resolution (*Zingoni, Taborri & Calabrò, 2024*). They can detect sensory, cognitive, or motor events by measuring EEG signals. The existing studies show that the DI has different event-related potentials in auditory perception and phonological processing (*Alqahtani, Alzahrani & Ramzan, 2023*; *Asvestopoulou et al., 2019*; *Zingoni, Taborri & Calabrò, 2024*). Neural processing abnormalities in preliterate children or newborns at risk for dyslexia may be assessed using EEG. By detecting brain indicators of dyslexia in the initial stages, therapies may be administered earlier, increasing the likelihood of long-term benefits (*Ahire et al., 2022*). Additionally, EEG can measure neural oscillations and cyclical patterns of electrical activity in the brain linked to various mental operations (*Elnakib et al., 2014*). These variations may play a role in issues with phonological processing and reading fluency.

Traditional dyslexia diagnosis techniques may be insensitive and less specific (*Kheyrkhah Shali & Setarehdan, 2020*; *Parmar & Paunwala, 2023*; *Yan, Zhou & Wong, 2022*). Standardized testing can ignore specific dyslexia symptoms, resulting in false-negative or positive findings. Due to the lack of early diagnosis, adults may face academic and psychological challenges (*Lou et al., 2017*). The majority of the current approaches to diagnosing dyslexia are based on subjective evaluations made by healthcare providers or educators, including the observation of reading problems or the administration of standardized tests. These subjective assessments may not correctly reflect dyslexia symptoms due to biases (*Deans et al., 2010*; *Tomaz Da Silva et al., 2021*). Typical methods for diagnosing dyslexia tend to focus on the more apparent symptoms, including challenges with reading or spelling. However, these significant symptoms may not reveal the full scope of dyslexia-related cognitive and neurological abnormalities (*Zingoni, Taborri & Calabrò, 2024*). Traditional techniques frequently employ static images of an individual for DD (*Banfi et al., 2021*; *Zainuddin et al., 2019*). As a developmental condition, dyslexia requires dynamic and longitudinal

examinations to track its course and responsiveness to treatments. Traditional dyslexia screening procedures may overlook cultural and language symptom variances (*Bernabini, Bonifacci & de Jong, 2021*; *Spoon, Crandall & Siek, 2019*). In different cultural and linguistic backgrounds, dyslexia may be misdiagnosed or underdiagnosed. Existing dyslexia screening techniques include reading or behavioral testing (*Ileri, Latifoğlu & Demirci, 2022*; *Psyridou et al., 2023*). The restricted emphasis may neglect critical information from brain imaging or cognitive testing, which helps better comprehend dyslexia (*Kariyawasam et al., 2019*; *Marimuthu, Shivappriya & Saroja, 2021*; *Vajs et al., 2022*).

DL models can extract complicated characteristics from EEG and fMRI data (*Asvestopoulou et al., 2019*; *Zingoni, Taborri & Calabrò, 2024*). These characteristics may include neural activity, brain region interaction, and neurological temporal dynamics. DL models can be trained to distinguish between dyslexic and non-dyslexic conditions using EEG and fMRI data (*Lr & Sudha Sadasivam, 2022*; *Parmar, Ramwala & Paunwala, 2021*; *Perera, Shiratuddin & Wong, 2018*). These models include convolutional neural networks (CNNs) for image-based FMRI data and recurrent neural networks (RNNs) for sequential EEG data (*Banfi et al., 2021*; *Frid & Manevitz, 2018*). By combining brain imaging data, these models may improve dyslexia diagnosis. DL models can reveal the brain mechanics of dyslexia, visualizing learned characteristics or constructing attention maps (*Jothi Prabha & Bhargavi, 2022*; *Zainuddin et al., 2016*). Researchers may use these visualizations to understand the role of unique brain regions in dyslexia and identify potential biomarkers. By integrating longitudinal EEG and FMRI data accumulated over time, DL models may be customized to individual patients (*Parmar, Ramwala & Paunwala, 2021*; *Perera, Shiratuddin & Wong, 2018*; *Tomaz Da Silva et al., 2021*). The existing techniques are not effective in detecting dyslexia in the early stages. There are few multi-modality-based DL models to identify dyslexia. In addition, the DL models demand extensive training and substantial computational resources to deliver optimal outcomes. The generalization of these models on real-time datasets is challenging and may produce false negative or positive results. Therefore, this study aimed to develop an automated model to detect dyslexia using multi-modality data. The study's contributions are listed below:

1. Enhanced MobileNet V3-model-based feature extraction using the SE block.
2. A self-attention mechanism-based EfficientNet B7 feature extraction model.
3. A self-attention mechanism and early stopping strategies based on the bi-directional long short term memory (Bi-LSTM) feature extraction model.
4. A hyper-parameter-tuned LightGBM-based DD detection model.
5. Generalization of the proposed DD detection model on diverse datasets.

The organization of this study is divided as follows: 'Literature Review' covers the existing literature on recent DL and artificial intelligence (AI) techniques for identifying dyslexia. The proposed methodology for extracting the crucial features and detecting dyslexia using multi-modal data is presented in 'Materials & Methods'. 'Results' provides the experimental setting and outcomes. The study contributions for DD are described in 'Discussion'. Lastly, 'Conclusions' concludes the proposed study.

# LITERATURE REVIEW

Research studies (*Ahire et al., 2022*; *Jan & Khan, 2023*; *Marimuthu, Shivappriya & Saroja, 2021*; *Perera, Shiratuddin & Wong, 2018*; *Usman et al., 2021*; *Velmurugan, 2023*) systematically reviewed DD techniques based on DL approaches. *Spoon, Crandall & Siek (2019)* used children's handwritten images to predict dyslexia. They employed a feature extraction technique to extract the crucial patterns to differentiate between normal and abnormal individuals. In order to identify cognitive processing variations associated with dyslexia, deep learning models may use behavioral data gathered during reading activities, including response times, accuracy rates, or eye-tracking patterns (*Deans et al., 2010*; *Nerušil et al., 2021*). These models may employ long short-term memory (LSTM) and transformer architectures for sequence modeling. To better understand dyslexia, deep learning methods, including multi-modal fusion networks and attention processes, can integrate data from multiple sources (*Lr & Sudha Sadasivam, 2022*). Transfer learning adapts pre-trained deep learning models to extract critical features for dyslexia diagnosis. By fine-tuning these models, researchers can increase detection accuracy with limited labeled data. To supplement limited DD datasets, generative adversarial networks may provide synthetic data samples replicating genuine FMRI and EEG data (*Lr & Sudha Sadasivam, 2022*; *Tomaz Da Silva et al., 2021*; *Yan, Zhou & Wong, 2022*). The generalizability and resilience of deep learning models for dyslexia detection may be enhanced using GANs by augmenting the variety and volume of the training data. *Zainuddin et al. (2018)* proposed a model using a K-nearest neighbor algorithm based on DDM. Similarly, studies (*Guhan Seshadri et al., 2023*; *Parmar & Paunwala, 2023*; *Yan, Zhou & Wong, 2022*) employed EEG signals for predicting dyslexia.

*Frid & Manevitz (2018)* discussed the importance of feature extraction techniques in DD. *Zaree, Mohebbi & Rostami (2023)* proposed an ensemble learning approach integrating multiple models' outcomes to predict dyslexia. *Raatikainen et al. (2021)* built a model to detect developmental dyslexia using the individual's eye movements. *Nerušil et al. (2021)* developed a model using eye movements. *Tomaz Da Silva et al. (2021)* proposed a visualization technique to identify DD using FMRI images. A CNN model was trained with high-level features of dyslexia. The studies (*Alqahtani, Alzahrani & Ramzan, 2023*; *Guhan Seshadri et al., 2023*; *Kariyawasam et al., 2019*) classified multiple medical imaging techniques for DD using CNN models. These studies extracted intricate patterns from EEG signals.

Handwritten and eye movement data are valuable in DD development (*Deans et al., 2010*; *Jothi Prabha & Bhargavi, 2022*; *Lou et al., 2017*; *Marimuthu, Shivappriya & Saroja, 2021*). However, the unique writing styles, character formation, and spatial structure of handwriting can cause complexities in identifying dyslexia. The interpretation of handwriting features is based on the evaluator's expertise. Thus, the ground truth information may vary from the annotated handwritten images. Similarly, extracting patterns from the eye movement data demands high computational resources. Reading, linguistic, cognitive, and visual processing speeds influence eye movement patterns. Eye movement patterns of dyslexic individuals may overlap with the patterns of normal

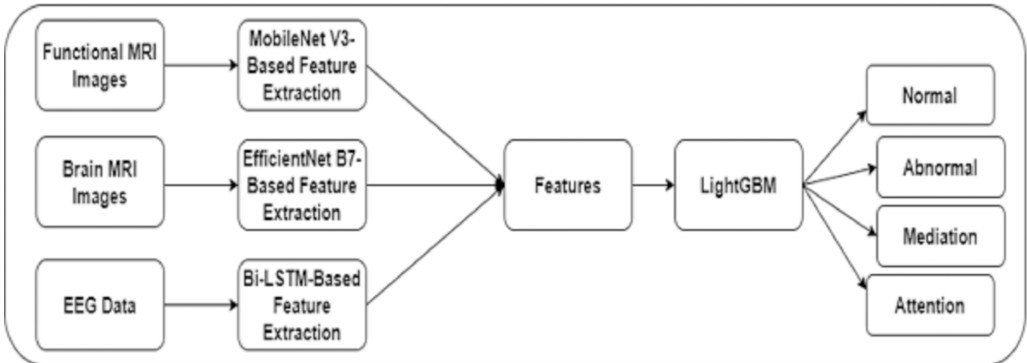

**Figure 1    The proposed framework.**

individuals. Individuals with dyslexia may exhibit compensating mechanisms or adaptive techniques during reading, altering or concealing their usual eye movement patterns, making dyslexia diagnosis more challenging. The interpretability of CNN and RNN-based DDM prediction is challenging for physicians in decision-making. The complex architecture of the existing models causes difficulties in deploying them in healthcare settings. Extensive data augmentation and transfer learning techniques are required to improve the efficiency of the existing models. In addition, the current models are susceptible to overfitting due to limited datasets. Developing a DL-based DDM from scratch may demand high-performance computing resources. The effective performance of the feature extraction plays a crucial role in lowering false negative/positive results. The recent developments in CNN and RNN architectures offer an opportunity to apply them in developing DDMs. These knowledge gaps have motivated the authors to focus on multi-modality-based DD detection models.

## MATERIALS & METHODS

The authors built a framework to identify dyslexia using multi-modality data. Initially, an image slicer generated images of size 224 × 224 from the FMRI images. The proposed framework included techniques for feature extraction using FMRI, MRI, and EEG data. In addition, a fine-tuned LightGBM model was used to predict dyslexia. Figure 1 shows the proposed framework for DD.

### Data acquisition

The authors acquired FMRI, MRI, and EEG data from three public repositories. Dataset 1 contained the raw 3 T MRI data and FMRI images of 58 children. The FMRI images were captured during the reading-related functional activities. Stimuli encompassing 60 words and 60 matched pseudo homophones were used to evaluate the individual's activity. The activities were repeated three times with a short break of 3–5 min. The dataset could be accessible using the repository (https://openneuro.org/datasets/ds003126/versions/1.1.0).

   Dataset 2 included the synthetic grayscale MRI images. These images were generated using the T1-weighted MRI images of 204 children. The dataset owners followed the FMRI

study (*Kuchinsky et al., 2012*) to develop the neuroimaging data. The adaptive non-local mean algorithm was used to denoise and correct the image biases. The Gaussian Kernel method was used to minimize the false positive results. The dataset could be downloaded from the repository (*Vaden et al., 2020*) (https://data.mendeley.com/datasets/3w9662wjpr/1).

Dataset 3 contained the EEG data of ten college students watching videos of introductory algebra and quantum mechanics. A total of 20 videos were used to collect data from the participants. A headband, MindSet was used to measure the participants' brain activity. The dataset covered 12,811 rows ×15 columns of EEG data, including subject ID, mediation, attention, delta, *etc.* The dataset was available in the Kaggle repository (https://www.kaggle.com/code/alrohit/eeg-analysis-of-confusion-for-dyslexia-diagnosis/notebook).

The datasets were available in the public repository. Researchers can use these data without any restrictions. The dataset owners obtained the participants' consent to share the data for research purposes.

In order to generate multiple images from the FMRI images, the authors developed a 3D CNN model. The model contained five convolution layers with filter sizes of 32, 64, 128, 256, and 512, and maxpooling layers. The features were flattened. Subsequently, dense and dropout layers with rectified linear unit (ReLu) and Sigmoid activation functions were employed to generate the images. A channel dimension integration and resize functions were applied to resize the images into $224 \times 224$ pixels. Moreover, data augmentation techniques were employed to train the feature extraction models.

## MobileNet V3-based feature extraction

The inverted residuals and architectural optimization improve the performance of the MobileNet V3 model in medical image classification and semantic segmentation. The utilization of the SE block, hard-swish activation function, and efficient last-stage convolution layers lead to better feature representation compared to the existing pre-trained models. The flexibility and adaptability of the MobileNet V3 model have motivated the authors to apply it in this study. In addition, the MobileNet V3 model was lightweight and supported the proposed DDM to detect dyslexia in a resource-constrained environment. The depthwise separable, convolutions, linear bottlenecks, and inverted residuals minimized the computational complexities. Channel-wise feature calibration was used to enhance the feature representation. However, integrating standard squeeze and excitation blocks could improve the model's efficacy.

The authors developed a shallow CNN model with a MobileNet V3 backbone in this study. They froze the initial layers of the MobileNet V3 model to prevent overfitting and reduce the computational cost. The lightweight nature of the MobileNet V3 model may reduce the ability to extract intricate features of dyslexia. Integrating the SE mechanism can improve the MobileNet model's performance by enhancing feature representation and discrimination power. In addition, the SE mechanism facilitates the MobileNet model in learning informative dyslexia features. Thus, the authors integrated the SE block with

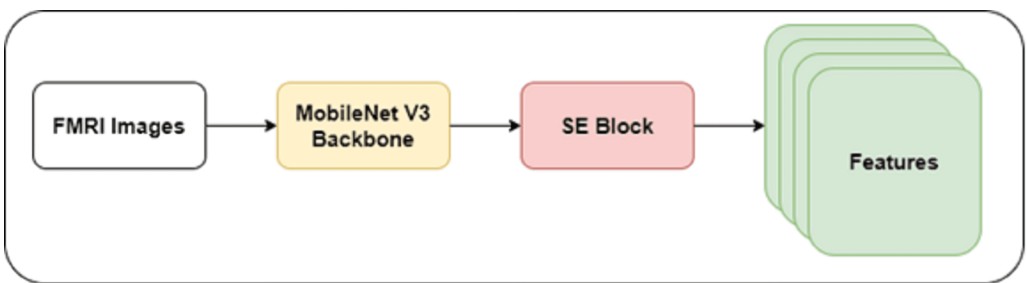

**Figure 2** **MobileNet V3-based feature extraction.**

the backbone model. Figure 2 highlights the architecture of the recommended feature extraction using the MobileNet V3-model.

Equation 1 shows the computational form of SE integration. Introducing the SE block into the model improved the adaptive recalibration of the channel-wise features. The SE block contained a global average pooling layer, reshape function, and dense layers.

$$T = MobileNetV3\_SE(X, F, K, E, S, R) \tag{1}$$

where T is the output tensor, X is the input tensor, F is the number of filters, K is the kernel size, E is the expansion ratio, S is the stride, and R is the squeeze ratio.

## EfficientNet B7-based feature extraction

The EfficientNet B7 model was the highly effective variant of the EfficientNet models. It offered complex patterns and features by maintaining computational efficiency. Medical imaging analysis models use the EfficientNet B7 models to generate a better outcome. EfficientNet model uses bottleneck layer and depthwise separable convolutions for feature extraction. The compound scaling functionality can balance model depth, width, and resolution. The EfficientNet model can enable high-quality, efficient, and scalable solutions for real-world challenges. These features of the EfficientNet B7 model motivated the authors to employ it for extracting key features from the FMRI images. The authors built a CNN model using six convolution layers, batch normalization, and dropout layers. They used the EfficientNet B7 model's weights to generate features. The compound scaling method was employed to maintain a trade-off between model depth and resolution. The self-attention mechanism was introduced using the global average pooling layer, reshape, and permute functions for the critical feature selection. The self-attention mechanism captures global dependencies in the multi-modality data, focusing on dyslexia-related patterns while neglecting irrelevant features. It assigns unique attention weights to different parts of the input sequences. It facilitates adaptive feature fusion across modalities. It identifies the structural differences in the brains of DIs compared to the normal individuals. These differences may encompass modifications in the size and asymmetry of brain regions related to language processing. The patterns associated with fractional anisotropy were extracted to assist the proposed model in producing optimal accuracy. Equation 2 shows

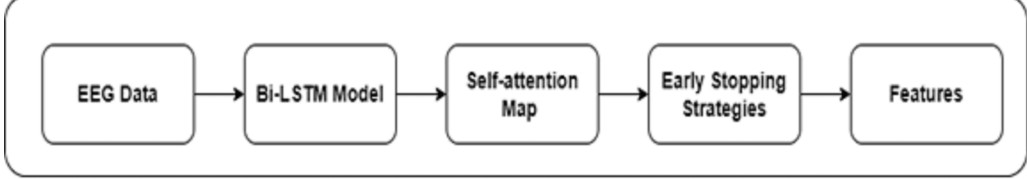

**Figure 3  Bi-LSTM-based feature extraction.**

the computation of the attention weights using a scaled dot-product attention approach.

$$attention\ weights = Softmax\left(Mat\_Mul\left(query, key, transpose\right)\right) \tag{2}$$

where the Mat_Mul function performs matrix multiplication of query and key tensors for the critical feature identification, transpose indicates the transposition of crucial tensor, and the Softmax function computes the exponential of each element of the matrix and normalizes it using the sum of all elements.

## Bi-LSTM-based feature extraction

The motivation to apply the Bi-LSTM model to extract features from EEG data stemmed from the bi-directional architecture. Bi-LSTM can capture temporal dependencies in sequential data. It can learn from past performance and tune itself for future direction. EEG data are susceptible to artifacts from eye movements, muscle activity, and external interference. Bi-LSTM can mitigate the effect of noise on feature extraction. To enhance the performance of the Bi-LSTM, the authors introduced a self-attention mechanism and early stopping strategies. In the context of EEG feature extraction, the self attention mechanism focuses on the mismatched negativity response, reflecting the brain's ability to identify and process auditory stimuli. The connectivity patterns among auditory, language, and reading-related brain regions were detected in order to predict dyslexia using EEG. The structure of Bi-LSTM-based feature extraction is highlighted in Fig. 3. Equation 3 shows the mathematical form of the Bi-LSTM layer.

$$P = Model.add\left(Bi-LSTM\left(Units = 64, return_{sequence} = True\right), input_{shape} = T, C\right) \tag{3}$$

where P is the prediction, Units specify the number of neurons, $return_{sequence}$ represents the sequence of outputs for each timestep, $input_{shape}$ shows the shape of input with time step for each sequence, and channels show the number of features.

The self-attention mechanism assists the Bi-LSTM model in extracting the key features of dyslexia. In addition, rectified linear unit (ReLu) and Softmax layers were integrated for DD prediction. The early stopping strategies were used to monitor the validation loss and improve the model's performance using the callback function to restore the best weights.

## Fine-tuned LightGBM-based DDM

LightGBM is a gradient-boosting technique widely applied for classification and regression. A histogram-based algorithm was used to compute gradients. Compared to the existing

gradient boosting technique, LightGBM produces a better result. It uses histogram-based algorithm for tree building, reducing the memory footprint. It supports out-of-core learning and distributed training that suits the proposed study to make predictions with minimum hardware configurations. In the context of DD, the class imbalance may cause challenges in identifying the optimal outcome. However, the inherent features, including weighted sampling and class-specific objectives, overcome the class imbalances. In addition, it can locate non-linear relationships between the features and the target variables. These advantages motivate the authors to employ the LightGBM model in this study.

Hyperband optimization was used to improve the performance of the LightGBM model. Using hyperparameter optimization, the authors fine-tuned the key parameters, including mm_leaves, max_depth, learning rate, n_estimator, reg_alpha, and reg_lambda. To control the step size during the boosting process, the authors set the ranges for learning rated from 0.01 to 0.1. An integer ranging from 3 to 10 was used for the maximum depth of each decision tree. The maximum number of trees per tree, ranging from 20 to 100, was used to optimize the LightGBM model. Feature fraction ranges from 0.5 to 1.0 were utilized for each split. The Hyperband algorithm evaluated the model using the randomly sampled hyper-parameter configuration. It iteratively evaluateed the performance and focused on resources associated with promising outcomes. The best-performing hyper-parameters configurations were selected based on the validation performance. Equation 4 shows the mathematical form of the fine-tuned LightGBM model.

$$Prediction = hyperband\left(LightGBM\left(n, m, l, n, l1, l2\right)\right) \tag{4}$$

where n is the maximum number of leaves, m is the maximum depth of each tree, l is the learning rate, n is the number of boosting rounds, $l1$ is the L1 regularization, and $l2$ is the L2 regularization.

### Evaluation metrics

To ensure the proposed model's ability to capture the crucial features of dyslexia, the authors applied multiple evaluation metrics. Accuracy ($Acc_y$) was used to measure the percentage of similarity between predicted and actual values. Precision ($Pre_c$) refers to the accuracy of the model's optimistic predictions. The higher value of precision indicates the absence of false positives. Recall ($Rec_l$) showed the percentage of true positives predicted by the proposed DD detection model. The existence of false negatives could be identified using the recall metric. F1-score integrated the outcomes of precision and recall. Cohen's Kappa (K) was used to evaluate the model's prediction in imbalanced datasets. In addition, the statistical significance of the model's outcomes was computed using standard deviation (SD) and confidence interval (CI).

## RESULTS

The authors implemented the proposed DDM using Windows 10 Professional, i7 14700K Processor, 16 GB RAM, and GeForce RTX 4060 Ti Eagle 8G environment. Table 1 presents the configuration settings of the MoblieNet V3, EfficientNet B7, Bi-LSTM, and LightGBM models. The datasets were divided into a train set (60%), a validation set (20%), and

**Table 1  Computational configurations.**

| Model | Parameters | Values |
|---|---|---|
| MobileNet V3 Backbone | Optimizer type | RMSProp |
| | Image Size | $224 \times 224$ |
| | Learning rate | $1 \times 10^{-4}$ |
| | Weight decay | $1 \times 10^{-5}$ |
| | Maximum epoch | 120 |
| Efficient B7 Backbone | Optimizer type | RMSProp |
| | Image Size | $224 \times 224$ |
| | Learning rate | $1 \times 10^{-5}$ |
| | Maximum epoch | 120 |
| LightGBM | Boosting type | Random forest |
| | Learning rate | $1 \times 10^{-4}$ |
| | Sub sample | 0.777 |
| | Regularization | L1 and L2 |
| | Optimization | Hyperband |

a test set (20%). The batch-wise training approach was followed to prevent overfitting. Based on the validation set, the authors fine-tuned the model's performance. PyTorch, Keras, TensorFlow, and Theano libraries were utilized for the model development. They employed pre-trained MobileNet V3 and DenseNet 201 models for the comparative analysis. The source codes of MobileNet V3, DenseNet 201, and SqueezeNet V1.1 models were extracted from the GitHub repositories. Figure 4 shows the proposed DD detection model performance in different batches. The model achieved an average accuracy of 97.2%, 98.1%, and 96.8% for datasets 1–3, respectively.

To overcome the model overfitting, the authors employed a validation set and Epoch-wise training approach. Figure 4 shows the performance of the proposed DDM in different Epochs. The model achieved an average accuracy of 98.1%, 98.2%, and 97.1% for datasets 1, 2, and 3.

Figure 5 highlights the training and validation loss at each epoch. Compared to the training loss, the validation loss was low. The recommended early-stopping strategy minimized the validation loss and achieved considerable accuracy. In addition, the self-attention mechanism and Hyperband optimization played a significant role in improving the efficiency of the proposed model.

Table 2 presents the performance of the proposed DDM. It reveals the significant improvement in the model performance. The proposed model identified the intricate patterns of dyslexia from the multiple modalities. The suggested feature extraction techniques yielded an exceptional outcome. The validation test fine-tuned the process of dyslexia identification. Figure 6 illustrates the findings of the proposed model.

The outcomes of the comparative analysis of models based on dataset 1 are presented in Table 3. The recommended MobileNet V3-based feature extraction supported the model in detecting dyslexia from the FMRI images. The image generation process produced high-quality images to train the MobileNet V3 model. In addition, introducing the SE

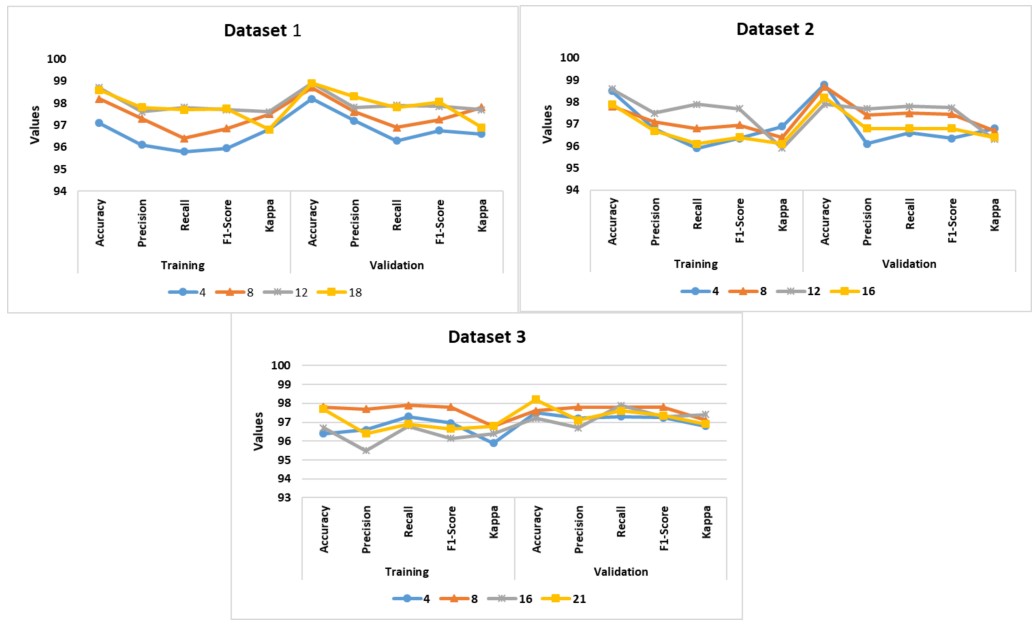

**Figure 4** Epoch-wise performance analysis.

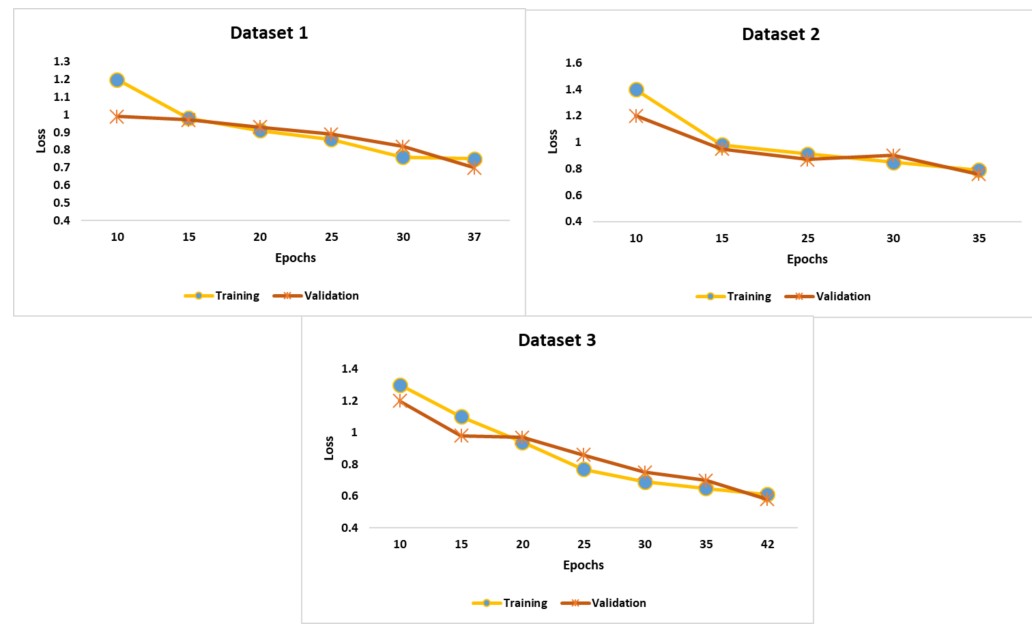

**Figure 5** Findings of performance analysis.

block assisted the proposed model in achieving optimal accuracy. The performance of the models is illustrated in Fig. 7.

The performance of the individual models based on dataset 2 is outlined in Table 4. It is evident that the suggested model outperformed the existing models. The feature

**Table 2  Performance evaluation outcomes.**

| Classes | $Acc_y$ | $Pre_c$ | $Rec_l$ | F1-Score | K |
|---|---|---|---|---|---|
| Normal | 98.7 | 97.6 | 96.8 | 97.2 | 96.7 |
| Abnormal | 98.9 | 97.4 | 97.1 | 97.2 | 95.8 |
| Mediation | 97.8 | 96.5 | 96.2 | 96.3 | 96.5 |
| Attention | 98.5 | 97.1 | 96.8 | 96.9 | 96.3 |
| **Average** | 98.5 | 97.2 | 96.7 | 96.9 | 96.3 |

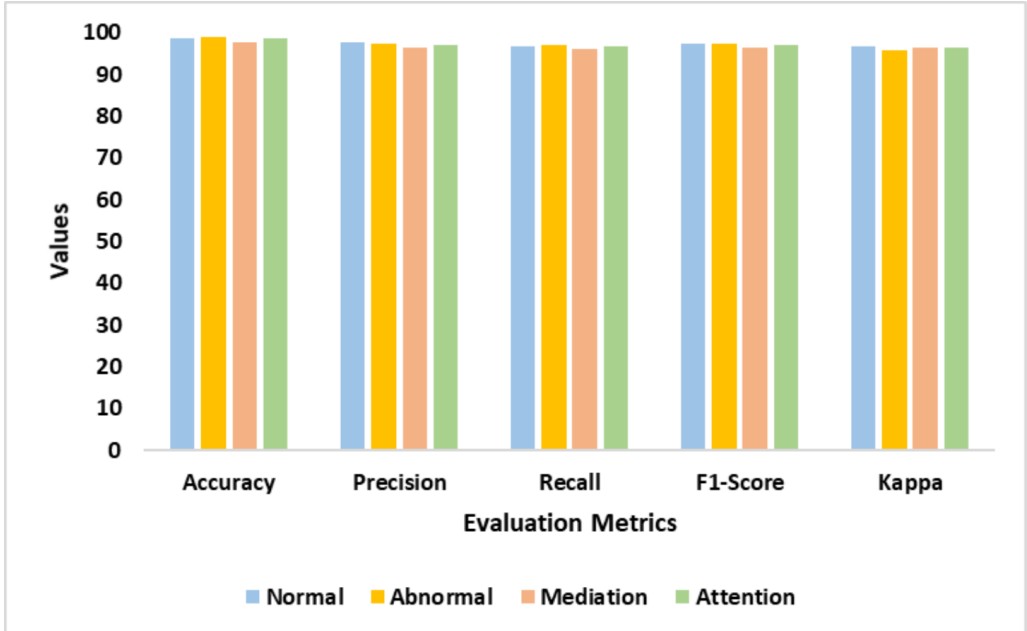

**Figure 6  Accuracy, precision, recall, F1-score, and Kappa represents the performance of the proposed model in detecting normal, abnormal, mediation, and attention classes.**

**Table 3  Comparative analysis –dataset 1.**

| Models | $Acc_y$ | $Pre_c$ | $Rec_l$ | F1-Score | K |
|---|---|---|---|---|---|
| *Lr & Sudha Sadasivam (2022)* | 97.2 | 96.1 | 96.7 | 96.4 | 95.7 |
| *Tomaz Da Silva et al. (2021)* | 94.7 | 94.5 | 94.4 | 94.4 | 93.2 |
| SqueezeNet V1.1 | 93.8 | 93.6 | 93.8 | 93.7 | 92.8 |
| MobileNet V3 | 96.5 | 95.1 | 95.3 | 95.2 | 94.3 |
| DenseNet 201 | 95.7 | 95.2 | 95.2 | 96.4 | 93.6 |
| EfficientNet B7 | 96.8 | 95.3 | 95.7 | 94.4 | 95.1 |
| Proposed model | 98.9 | 96.6 | 97.1 | 96.8 | 96.7 |

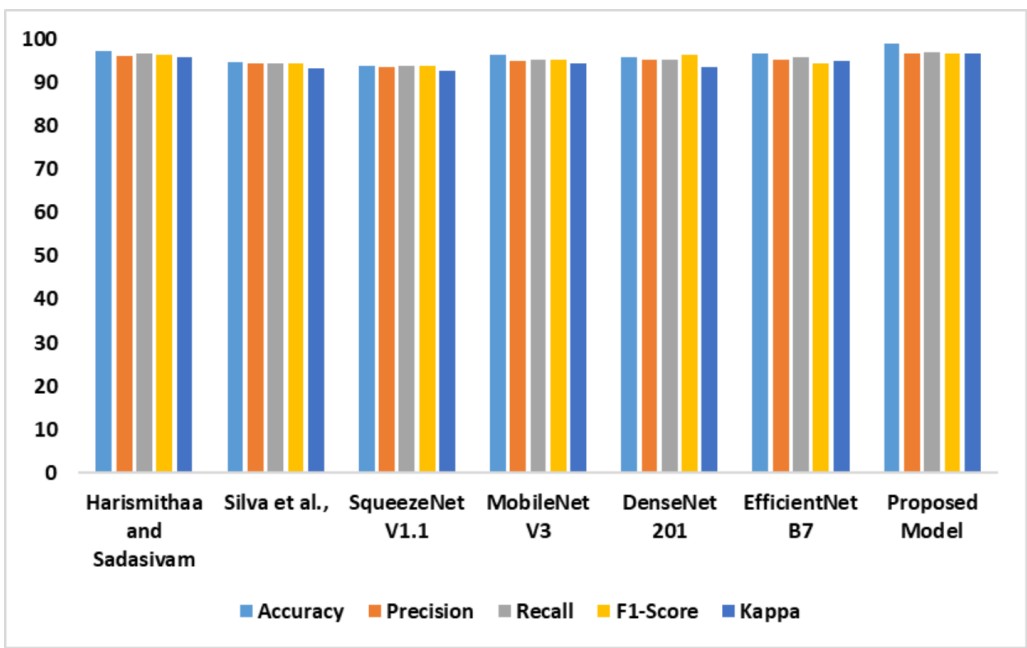

**Figure 7** Accuracy, precision, recall, F1-score, Kappa indicates the performance of the dyslexia detection models on dataset 1.

**Table 4** Comparative analysis –dataset 2.

| Models | Acc$_y$ | Pre$_c$ | Rec$_l$ | F1-Score | K |
|---|---|---|---|---|---|
| *Lr & Sudha Sadasivam (2022)* | 95.8 | 95.1 | 95.3 | 95.2 | 94.1 |
| SqueezeNet V1.1 | 96.1 | 96.4 | 96.6 | 96.5 | 93.8 |
| MobileNet V3 | 96.7 | 95.9 | 95.7 | 95.8 | 93.9 |
| DenseNet 201 | 95.5 | 95.1 | 94.9 | 95.0 | 94.7 |
| EfficientNet B7 | 96.6 | 96.2 | 95.8 | 96.0 | 95.5 |
| Proposed model | 98.6 | 96.5 | 96.8 | 96.6 | 95.8 |

extraction and self-attention mechanisms improved the performance of the proposed model. Compared with the standard EfficientNet B7, the fine-tuned Efficient B7 with a self-attention mechanism yielded a better result. Figure 8 represents the findings of the comparative analysis.

The capability of the proposed model in identifying dyslexia using EEG data is presented in Table 5. There is a significant improvement in the proposed model's performance compared with the baseline models. The outcomes of the comparative analysis are outlined in Fig. 9.

The statistical significance of the outcomes is presented in Table 6. The findings revealed the reliability of the proposed model. It suggested that the model could be implemented in healthcare settings.

The computational strategies highlighted the computational resource demands for the models to detect dyslexia. The findings indicated that the proposed model required less

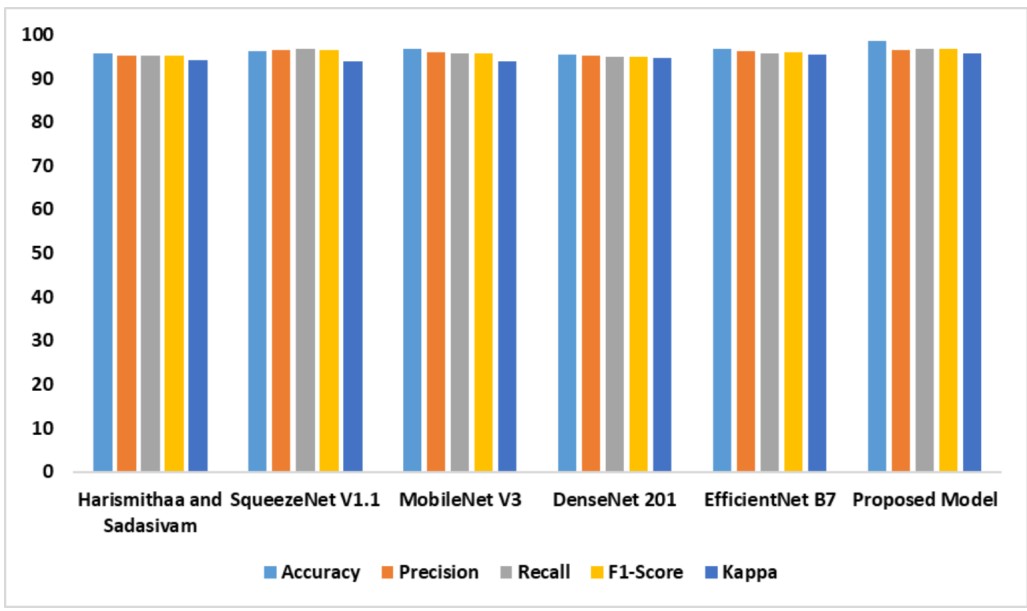

**Figure 8** Accuracy, precision, recall, F1-score, Kappa indicates the performance of the dyslexia detection models on dataset 2.

**Table 5** Comparative analysis –dataset 3.

| Models | $Acc_y$ | $Pre_c$ | $Rec_l$ | F1-Score | K |
|---|---|---|---|---|---|
| *Christodoulides et al. (2022)* | 94.1 | 93.1 | 93.3 | 93.2 | 91.2 |
| Bi-LSTM model | 97.1 | 96.6 | 96.8 | 96.7 | 93.6 |
| *Parmar & Paunwala, (2023)* | 96.7 | 95.8 | 94.7 | 95.2 | 93.7 |
| *Guhan Seshadri et al. (2023)* | 97.5 | 97.1 | 97.3 | 97.2 | 96.5 |
| Proposed model | 98.8 | 98.6 | 98.3 | 98.4 | 96.3 |

computational power to generate an exceptional outcome. It is evident that the proposed model could be implemented in a resource-constrained environment. Table 7 presents the computational complexities of the models.

## DISCUSSION

A multi-modal-based DL model was introduced to detect dyslexia. The authors trained the model using FMRI, MRI, and EEG data. They constructed an image slicer using 3D CNN to extract images from the FMRI images. A CNN model was developed using the weights of the MobileNet V3 model. In addition, the SE block was integrated with the model to extract the crucial features of dyslexia. Likewise, using the EfficientNet B7 as a backbone, a feature extraction model was developed to extract features from the MRI images. The model was fine-tuned using a self-attention mechanism. To extract the feature from EEG data, the authors used the Bi-LSTM model. Self-attention mechanisms and early stopping strategies were used to fine-tune the model. Finally, the LightGBM model was used to

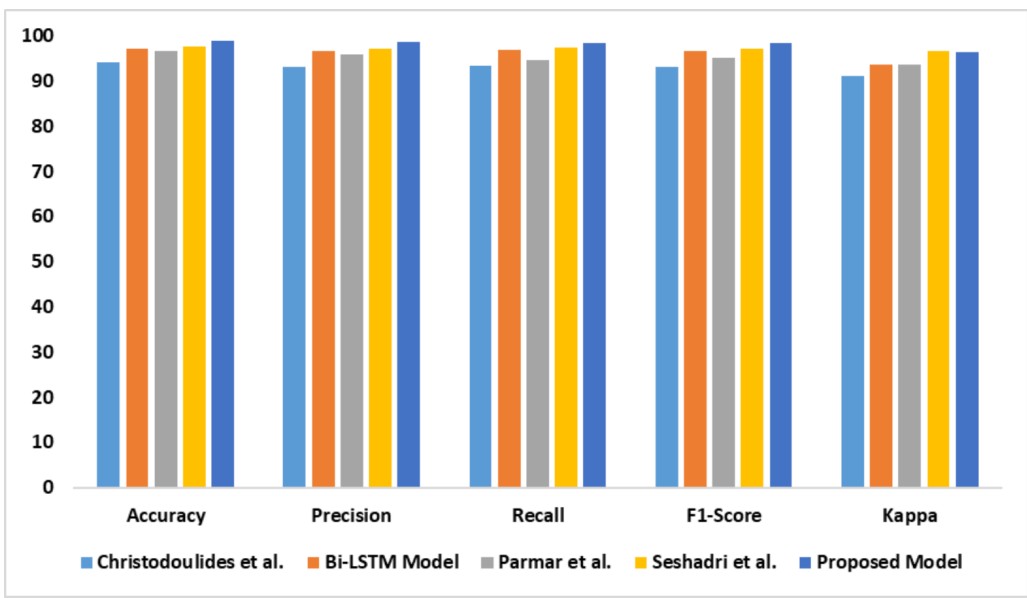

**Figure 9  Accuracy, precision, recall, F1-score, Kappa indicates the performance of the dyslexia detection models on dataset 3.**

**Table 6  Statistical analysis.**

| Dataset | Models | SD | CI | Loss |
|---|---|---|---|---|
| Dataset 1 | *Lr & Sudha Sadasivam (2022)* | 0.0005 | 95.6 | 0.54 |
| | *Tomaz Da Silva et al., (2021)* | 0.0004 | 96.2 | 0.75 |
| | SqueezeNet V1.1 | 0.0005 | 95.3 | 0.36 |
| | MobileNet V3 | 0.0004 | 96.4 | 0.89 |
| | DenseNet 201 | 0.0004 | 95.2 | 1.12 |
| | EfficientNet B7 | 0.0004 | 95.5 | 0.79 |
| | Proposed model | 0.0003 | 96.1 | 0.41 |
| Dataset 2 | *Lr & Sudha Sadasivam (2022)* | 0.0005 | 95.3 | 0.48 |
| | SqueezeNet V1.1 | 0.0003 | 95.6 | 0.74 |
| | MobileNet V3 | 0.0004 | 96.1 | 0.77 |
| | DenseNet 201 | 0.0005 | 95.1 | 1.14 |
| | EfficientNet B7 | 0.0004 | 95.3 | 0.54 |
| | Proposed model | 0.0004 | 95.8 | 0.39 |
| Dataset 3 | *Christodoulides et al. (2022)* | 0.0005 | 95.5 | 0.56 |
| | Bi-LSTM model | 0.0004 | 95.3 | 0.71 |
| | *Parmar & Paunwala, (2023)* | 0.0003 | 95.1 | 0.48 |
| | *Guhan Seshadri et al. (2023)* | 0.0004 | 95.8 | 0.89 |
| | Proposed model | 0.0004 | 96.1 | 0.44 |

**Table 7  Computational complexities.**

| Dataset | Models | Parameters (in Millions) | Flops (in Giga) | Testing time (in Seconds) | Learning rate |
|---|---|---|---|---|---|
| Dataset 1 | *Lr & Sudha Sadasivam (2022)* | 36 | 39 | 0.357 | $1 \times 10^{-4}$ |
| | *Tomaz Da Silva et al. (2021)* | 42 | 29 | 0.453 | $1 \times 10^{-3}$ |
| | SqueezeNet V1.1 | 45 | 33 | 0.324 | $1 \times 10^{-3}$ |
| | MobileNet V3 | 37 | 31 | 0.335 | $1 \times 10^{-4}$ |
| | DenseNet 201 | 53 | 43 | 0.463 | $1 \times 10^{-3}$ |
| | EfficientNet B7 | 41 | 33 | 0.518 | $1 \times 10^{-4}$ |
| | Proposed model | 31 | 29 | 0.277 | $1 \times 10^{-5}$ |
| Dataset 2 | *Lr & Sudha Sadasivam (2022)* | 39 | 36 | 0.413 | $1 \times 10^{-4}$ |
| | SqueezeNet V1.1 | 48 | 34 | 0.313 | $1 \times 10^{-3}$ |
| | MobileNet V3 | 39 | 33 | 0.327 | $1 \times 10^{-4}$ |
| | DenseNet 201 | 55 | 44 | 0.469 | $1 \times 10^{-2}$ |
| | EfficientNet B7 | 43 | 34 | 0.504 | $1 \times 10^{-4}$ |
| | Proposed model | 30 | 27 | 0.316 | $1 \times 10^{-5}$ |
| Dataset 3 | *Christodoulides et al. (2022)* | 53 | 42 | 0.714 | $1 \times 10^{-2}$ |
| | Bi-LSTM model | 43 | 54 | 0.426 | $1 \times 10^{-3}$ |
| | *Parmar & Paunwala, (2023)* | 54 | 44 | 0.589 | $1 \times 10^{-2}$ |
| | *Guhan Seshadri et al. (2023)* | 47 | 51 | 0.631 | $1 \times 10^{-2}$ |
| | Proposed model | 39 | 42 | 0.323 | $1 \times 10^{-4}$ |

detect dyslexia using the extracted features. The model was generalized on three datasets. The exceptional performance of the proposed model is highlighted in Tables 3–4. The suggested feature extraction techniques supported the fine-tuned LightGBM models to identify dyslexia with limited computational power.

By implementing the proposed model in educational centers, teachers can identify dyslexia using wearable EEG devices. Each modality illuminates dyslexia biomarkers and characteristics differently. By integrating data from multiple modalities, the proposed model improves prediction accuracy, generalization, interpretability, and individualized intervention techniques for dyslexia detection and diagnosis. It detects intricate patterns and correlations in data using several modalities, boosting prediction accuracy and diagnostic performance. It can be used to personalize therapies and support services for dyslexic individuals by considering their unique genetic profiles, brain structure and function, cognitive ability, and environmental circumstances.

In line with *Tomaz Da Silva et al.*'s (*2021*) findings, the proposed model employed FMRI images and generated a remarkable outcome. They used 3D CNN to extract dyslexia patterns from the FMRI data. The 3D CNN required a substantial set of parameters and operators, leading to longer training time. It demands a larger memory footprint compared to the proposed DDM. In addition, real-time processing of 3D volumes may demand specialized hardware accelerators. The suggested feature extraction techniques supported the proposed model to overcome the challenges in DD. Compared with the *Lr & Sudha Sadasivam (2022)*, the proposed model achieved a better result. Moreover, it requires less computational power. *Lr & Sudha Sadasivam (2022)* employed time

distribution convolutional LSTM for feature extraction. The recurrent layers of this model can reduce parallelism during training and inference. The proposed model followed a similar approach to the *Christodoulides et al. (2022)* model for extracting features from FMRI and MRI data. In addition, the proposed model utilized the EEG data to identify dyslexic individuals. *Parmar & Paunwala (2023)* used a predictor extraction and selection methodology to predict dyslexia using EEG data. The shortcomings of their feature extraction reduced the performance of dyslexia identification. In contrast, the fine-tuned Bi-LSTM-based feature extraction assisted the proposed model in classifying the EEG data. *Guhan Seshadri et al. (2023)* applied a shallow CNN model for DD. Likewise, the proposed model employed shallow CNN models for extracting features from multi-modality data. The Bi-GRU model is complex compared to the suggested DDM. The model's complexity can lead to substantial training time and higher computational power. The existing models require additional dependencies between forward and backward passes, causing difficulties in propagating gradients effectively. The proposed model outperformed the pre-trained models, including MobileNet V3, SqueezeNet V1.1, DenseNet 201, and EfficientNet B7. These models demand high computational resources to classify the images.

The authors utilized public repositories to train and test the proposed DDM. EEG and MRI data cover sensitive information related to the individual's brain activities. The authors conducted rigorous validation to assess the diagnostic accuracy, sensitivity, specificity, and reliability of the proposed model. The proposed model generated the outcome without revealing the individual's sensitive data. The dataset owners obtained consent from the participants before collecting EEG and MRI data. In addition, they protected the individual's personal information by revealing EEG and MRI data without the individual's personal information. In the clinical setting, healthcare centers should follow robust data protection measures, including access controls and anonymization techniques to ensure individual data privacy and confidentiality.

The authors faced challenges in extracting features from multiple sources. The variability in the image quality caused challenges to the feature extraction. The authors developed the image slicer to extract images from various angles and normalize the image qualities. In addition, the data augmentation technique was required to increase the dataset size and overcome the class imbalance. The MobileNet V3 backbone demanded additional resources to extract the crucial features. The authors reduced the computation time by integrating the SE block with the MobileNet V3 model. Similarly, the performance of the EfficientNet B7-based feature extraction model was improved by introducing the self-attention mechanism. Addressing the proposed model's limitations can improve its interpretability and generalization in a real-time environment. The proposed model was trained using synthetic MRI images. It demanded substantial training to generate effective outcomes with novel MRI images. An effective wearable EEG device is required to track the behaviors of individuals in healthcare settings.

In the future, the proposed model can be trained with additional modalities in order to cover wide age groups. The authors will employ vision transformers to integrate the interpretability of the proposed model's outcomes. Vision transformers have an inherent attention mechanism, capturing the relationship between the multiple regions of FMRI

images. By utilizing class activation maps, healthcare professionals can visualize the key features and understand the decision logic. The layerwise analysis features can offer a deeper understanding of the proposed model's internal process.

## CONCLUSIONS

In this study, a DL-based model for detecting dyslexia using multi-modality data was proposed. The authors employed feature engineering techniques to identify the crucial patterns of dyslexia. The feature engineering comprised fine-tuned MobileNet V3, EfficientNet B7, and Bi-LSTM models. The authors integrated the SE block with the MobileNet V3 model to generate key features from FMRI images. They introduced a self-attention mechanism in EfficientNet B7 and Bi-LSTM to reduce the training time. The hyper-parameters of the LightGBM model were optimized using hyperband optimization. The proposed model was generalized on three datasets. The findings supported the exceptional performance of the proposed DD detection model. The recommended model outperformed the state-of-the-art DDMs with limited computational power. Healthcare centers can benefit from the suggested model. The multi-modality feature allows healthcare centers to detect dyslexia using FMRI images and EEG data. However, the proposed model faced challenges in integrating features from the CNN models and fine-tuning the performance of the LightGBM model. The proposed model demands substantial training in order to enhance its performance. The performance of the proposed model can be improved with vision transformers and graph neural networks.

### Funding

The King Salman Center for Disability Research funded this work through Research Group no KSRG-2022-079. The funders had no role in study design, data collection and analysis, decision to publish, or preparation of the manuscript.

### Grant Disclosures

The following grant information was disclosed by the authors:
The King Salman Center for Disability Research: KSRG-2022-079.

### Competing Interests

The authors declare there are no competing interests.

### Author Contributions

- Yazeed Alkhurayyif conceived and designed the experiments, performed the experiments, analyzed the data, performed the computation work, prepared figures and/or tables, authored or reviewed drafts of the article, and approved the final draft.
- Abdul Rahaman Wahab Sait conceived and designed the experiments, performed the experiments, analyzed the data, performed the computation work, prepared figures and/or tables, authored or reviewed drafts of the article, and approved the final draft.

## Data Availability

The FMRI dataset is available at: Chiara Banfi and Karl Koschutnig and Kristina Moll and Gerd Schulte-Koerne and Andreas Fink and Karin Landerl (2021). MRI Lab Graz: Reading-related functional activity in children with isolated spelling deficits and dyslexia. OpenNeuro. [Dataset] doi: 10.18112/openneuro.ds003126.v1.1.0.

The MRI dataset is available at Mendeley: *Vaden et al. (2020)*, "Data for: A principled approach to synthesize neuroimaging data for replication and exploration", Mendeley Data, V1, doi: 10.17632/3w9662wjpr.1.

The EEG dataset is available at Kaggle: https://www.kaggle.com/code/alrohit/eeg-analysis-of-confusion-for-dyslexia-diagnosis/notebook, accessed on 11, May 2023.

The code is available in the Supplemental Files.

## Supplemental Information

Supplemental information for this article can be found online at http://dx.doi.org/10.7717/peerj-cs.2077#supplemental-information.

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
