# Peer review of "Deep learning-driven dyslexia detection model using multi-modality data"

_PeerJ Computer Science, doi:10.7717/peerj-cs.2077_

## Round 0.1 · original submission · Minor Revisions

Please address all the comments.

**Language Note:** The review process has identified that the English language must be improved. PeerJ can provide language editing services - please contact us at [email protected] for pricing (be sure to provide your manuscript number and title). Alternatively, you should make your own arrangements to improve the language quality and provide details in your response letter. – PeerJ Staff

Reviewer 1 ·

Basic reporting

1. The author's proposed Hybrid Dyslexia detection model using multi modality data. The work is good but some of the abbreviations like SE, DD were not clear from the text and what does that actually signifies.
2. There are minor grammatical errors throughout the whole manuscript.

Experimental design

The justification of why author's have used Light GBM and mobile Vnet v3 model is not clear. In literature there are many models apart from Light GBM also who can produce better results. If Light GBM and mobile Vnet V3 is good then provide proper justification why author's have used only these models and why not other models.

Validity of the findings

Clearly describe the Loss obtained while running each epoch in all the three datasets.

Additional comments

The paper needs minor revisions before acceptance.

Cite this review as

Reviewer 2 ·

Basic reporting

The manuscript thoroughly explores a proposed deep learning model for identifying dyslexia, leveraging multi-modal data such as EEG and FMRI.
The introduction skillfully sets the stage for the research, offering a thorough background and rationale. However, elaborating more on why specific deep learning architectures and methodologies were chosen would make the paper clearer.
Additionally, providing additional details on technical aspects like hyperparameter optimization and the incorporation of self-attention mechanisms would enhance transparency.

Experimental design

The authors lay out their experiment details clearly, covering everything from the datasets they used to how they measured their model's performance.
Using multiple evaluation metrics is a plus, but they should provide more information on how they tuned their model's parameters and how they processed their data.

Validity of the findings

The results show promise for detecting dyslexia accurately, especially since they used various evaluation methods.
Still, there are some concerns about transparency, like how they used self-attention mechanisms and handled their dataset.
Addressing these concerns would make their findings more trustworthy.

Additional comments

The paper introduces an interesting way to detect dyslexia using deep learning and different types of data. It is great that they used advanced techniques and thoroughly evaluated their model, but they need to explain certain parts better to improve the overall quality of their work.

Scope for Improvement:
Justification of Model Selection: It would help to explain why they chose specific deep learning methods for analyzing EEG and FMRI data in more detail.
Transparency in Hyperparameter Optimization: Providing more info on how they adjusted their model's parameters and how they optimized them would make their study more reliable.
Clarity on Self-Attention Mechanisms: They need to explain how using self-attention mechanisms affected their model's performance in simpler terms.
Addressing Potential Biases: Discussing how they dealt with biases in their data or model training process would make their findings more reliable.
Comparison with Baseline Models: It would be helpful to compare their model to other existing methods in dyslexia detection to see how well it performs in comparison.

Cite this review as

Reviewer 3 ·

Basic reporting

The article presents a comprehensive study on the development of a deep learning-based Dyslexia Detection Model (DDM) using multi-modality data, including magnetic resonance imaging (MRI), functional MRI (fMRI), and electroencephalography (EEG). The integration of advanced deep learning techniques such as squeeze and excitation (SE) integrated MobileNet V3, self-attention mechanisms (SA) based EfficientNet B7, and early stopping and SA-based Bi-directional Long Short-Term Memory (Bi-LSTM) models, alongside the application of the LightGBM model fine-tuned with Hyperband optimization, marks a significant step forward in the early detection of dyslexia. The achieved accuracies (98.9% for fMRI, 98.6% for MRI, and 98.8% for EEG datasets) highlight the model's effectiveness. Despite the clear strengths of the study, a few areas could benefit from minor improvements or further elaboration:

While the performance metrics are impressive, the paper could benefit from a more detailed description of the datasets used, including their size, demographic information, and the criteria for inclusion or exclusion of participants. Clarifying whether these datasets are publicly available or can be accessed by other researchers for replication purposes would also be beneficial.

Experimental design

The suggestion to integrate vision transformers for enhanced feature extraction is a valuable direction for future research, especially in improving model interpretability. It would be helpful if the authors could elaborate on the current limitations regarding interpretability and how vision transformers might specifically address these issues. Additionally, examples of how interpretability could impact clinical or educational interventions would make a compelling case for its importance.

The paper states that the proposed model outperforms existing DDMs but does not provide a detailed comparison. A more comprehensive comparative analysis, including specific models that were outperformed, the metrics used for comparison, and the context in which these models were evaluated, would strengthen the claims of superiority.

Validity of the findings

The paper should discuss how well the proposed DDM generalizes across different age groups, languages, and educational backgrounds. Dyslexia manifests differently across individuals, so understanding the model's performance across diverse populations is crucial for its applicability in real-world scenarios.

While the paper briefly mentions future directions, a more detailed discussion on the limitations of the current study would be valuable. For instance, how does the model perform with noisy data or in real-time applications? What are the computational requirements for implementing this model in low-resource settings? Addressing these questions can provide a more rounded view of the research and guide future studies.

Additional comments

Finally, incorporating a section on ethical considerations, especially concerning privacy, consent, and the potential for misdiagnosis, would be pertinent. Given the sensitivity of medical data and the implications of a dyslexia diagnosis, discussing these aspects is critical for a holistic understanding of the model's impact.

Cite this review as

---

## Round 0.2 · accepted · Accept

After carefully reviewed the revised manuscript and I am pleased to confirm that all of the reviewers' comments have been adequately addressed. The revisions have greatly improved the clarity and quality of the manuscript. Since the previous reviewers were not invited to reevaluate the revised version, I took it upon myself to assess the revisions, and I am happy to report that the current version meets the standards for publication. Congratulations!